# Application of the ARIMA Model to Predict Under-Reporting of New Cases of Hansen’s Disease during the COVID-19 Pandemic in a Municipality of the Amazon Region

**DOI:** 10.3390/ijerph19010415

**Published:** 2021-12-31

**Authors:** Valéria Perim da Cunha, Glenda Michele Botelho, Ary Henrique Morais de Oliveira, Lorena Dias Monteiro, David Gabriel de Barros Franco, Rafael da Costa Silva

**Affiliations:** 1Graduate Program in Intellectual Property and Information Technology Transfer - PROFNIT, Federal University of Tocantins, Palmas 77001-090, TO, Brazil; glendabotelho@uft.edu.br (G.M.B.); aryhenrique@mail.uft.edu.br (A.H.M.d.O.); 2Medicine Course, State University of Tocantins, Palmas 77020-122, TO, Brazil; lorena.dm@unitins.br; 3Graduate Program in Digital Agroenergy (PPGADIGITAL), Federal University of Tocantins, Palmas 77001-090, TO, Brazil; david.franco@uft.edu.br; 4Computing Department, Federal University of São Carlos, São Carlos 13565-905, SP, Brazil; rafaelcs@estudante.ufscar.br

**Keywords:** leprosy, neglected disease, COVID-19, time series analysis, epidemiology

## Abstract

This work aimed to apply the ARIMA model to predict the under-reporting of new Hansen’s disease cases during the COVID-19 pandemic in Palmas, Tocantins, Brazil. This is an ecological time series study of Hansen’s disease indicators in the city of Palmas between 2001 and 2020 using the autoregressive integrated moving averages method. Data from the Notifiable Injuries Information System and population estimates from the Brazilian Institute of Geography and Statistics were collected. A total of 7035 new reported cases of Hansen’s disease were analyzed. The ARIMA model (4,0,3) presented the lowest values for the two tested information criteria and was the one that best fit the data, as AIC = 431.30 and BIC = 462.28, using a statistical significance level of 0.05 and showing the differences between the predicted values and those recorded in the notifications, indicating a large number of under-reporting of Hansen’s disease new cases during the period from April to December 2020. The ARIMA model reported that 177% of new cases of Hansen’s disease were not reported in Palmas during the period of the COVID-19 pandemic in 2020. This study shows the need for the municipal control program to undertake immediate actions in terms of actively searching for cases and reducing their hidden prevalence.

## 1. Introduction

Hansen’s disease still represents a serious public health problem in some countries around the world. Despite the reduction in the detection coefficients of new cases, predominantly after the implementation of multidrug therapy, there are countries that still have a high burden of the disease, including Brazil. It is an infectious–contagious disease with a slow evolution which mainly affects the skin and peripheral nerves, has an essentially clinical and epidemiological diagnosis and if not treated early, it can leave irreversible sequelae [1,2,3].

Brazil is highly endemic for Hansen’s disease, except in the south and southeast regions which maintain a threshold considered average by the Ministry of Health. The country ranks second in the detection of new cases in the world and demonstrates a heterogeneous distribution. In 2019, the Brazilian states with the highest overall detection coefficients were Mato Grosso and Tocantins, with 129.38 and 96.44 new cases per 100,000 inhabitants, respectively. In 2020, the same states continued to lead this ranking with detection rates of 71.44 and 53.95, respectively. In the city of Palmas, capital of Tocantins, the indicator of the detection rate of new cases for 100,000 inhabitants was 226.99 in 2019, which was the highest among the state capitals of the country [4,5].

In 2016, the Palmas Free from Hansen’s Disease Project was implemented in the capital of the state of Tocantins, which found, after training in primary care units for the recognition and treatment of the disease, a delay in the diagnosis performed by the health service. This was evidenced with an increase, after the implementation of the project, of 111.1% in the detection coefficient in children under 15 years of age, 201.1% in case detection by contact assessment, and 104.6% in the detection coefficient of new cases. All this effort demonstrated failures in the diagnosis of the disease and restructured the health service to obtain better performance [6].

In March 2020, the first case of COVID-19 was registered in Palmas and as in the rest of the world, confirmed cases resulting from this disease grew tremendously, reaching more than 20,000 confirmed cases in December 2020, as shown in Figure 1 [7,8].

Given this unusual reality, the vulnerability of the health system became more evident. In addition to the lack of knowledge about the pandemic, there were still challenges related to changes in behavior such as wearing a mask and social isolation, affecting the population’s mental health as well as physical health. Great challenges were imposed, especially for disadvantaged people who already lived in precarious conditions of life and health, which accentuated difficulties for people simultaneously affected by COVID-19 and other communicable or non-communicable diseases [9].

In Palmas, at the end of the first year of the pandemic, there was a decrease in the number of new reported cases of leprosy. According to Cruz [10], expectations regarding Hansen’s disease in the post-pandemic scenario are not optimistic. The existing difficulties in terms of access to and healthcare and healthcare information are aggravated and, along with the cancellation of early diagnosis activities, suspension of care and some specific problems in the provision of multidrug therapy (MDT), in addition to factors related to the vulnerabilities of these patients, can cause a huge setback in efforts to reduce the burden of this disease in the country [11,12].

In this scenario, the prediction of disease behavior becomes extremely important for epidemiological analyses and the planning of actions and public policies in the interest of public health, especially considering the influence of an external factor as relevant as a pandemic. Thus, this study aimed to apply the ARIMA model to predict the under-reporting of new Hansen’s disease cases during the COVID-19 pandemic in Palmas, Tocantins. This predictive model intends to measure the number of undiagnosed cases, and with this, it intends to assist the actions of epidemiological surveillance, providing information on the incidence of the number of leprosy cases as a result of the pandemic, helping public health managers in decision making.

## 2. Materials and Methods

### 2.1. Study Area

This study was performed in the city of Palmas, the state capital of Tocantins, located in the northern region of Brazil. Palmas is the country’s newest capital, whose estimated population in 2020 was approximately 306,000 inhabitants [13]. The Health Care and Surveillance Network (known in Brazil by the Portuguese acronym, RAVS) of the Health Department of Palmas was established by Ordinance No. 457/2019. RAVS has 3 administrative districts subdivided into 8 health territories with their respective 34 community health centers and other service spots. The care network has 85 family health strategy units (ESFs), 75 oral health teams, 506 community health agents (CHAs) and 13 expanded family health care centers (known in Brazil by the Portuguese acronym, NASF) and 1 street clinic team [14].

### 2.2. Data Source

The time series study was fulfilled with information from compulsory notifications of Hansen’s disease in the city of Palmas during the period between January 2001 and December 2020. Data were obtained from notifications of new cases registered in the Notifiable Diseases Information System (known in Brazil by the Portuguese acronym, SINAN). Cases with a “diagnostic error” in the exit criterion attribute (130 records) were excluded. The source for the population data was the Brazilian Institute of Geography and Statistics [13] (known in Brazil by the Portuguese acronym, IBGE), based on the population census and the population estimate in the inter-census years.

### 2.3. Exploratory Analysis

Data were received in electronic spreadsheet format, and from there, the cleaning and selection of attributes for the research, exploratory analysis and calculation of indicators were performed according to the metrics of the Ministry of Health. The proportions of notifications by sex and by operational classification, which indicates late diagnosis, were calculated and evaluated for this study in addition to the following indicators:Annual detection coefficient of new cases per 100,000 inhabitants, which assesses the magnitude of the disease and estimates the risk of the occurrence of new cases;Annual detection coefficient of new cases in people aged 0–14 years per 100,000 people, which measures the strength of recent transmission;Proportion of cases with grade II physical disability among the new cases detected in the year, which estimates the capacity for the early detection and the hidden endemic;Proportion of cured cases among new cases in the cohort years, which assesses the quality of attendance and treatment effectiveness.

These indicators were analyzed during the years 2001–2020, using the trend of the time series in order to understand the behavior of indicators influenced by the external factor—the COVID-19 pandemic.

### 2.4. Predictive Model

At the same time, the number of registered notifications was used, grouped by month/year, to create a predictive model through the analysis of the time series. In these models, the smaller the number of records available, the greater the confidence interval, and this was the reason why we chose to use the raw data grouped together. The statistical chosen model was the autoregressive integrated moving average (ARIMA), which is the most frequently used model for the analysis of time series in health [15,16,17]. The ARIMA (p,D,q) process generates nonstationary series that are integrated of order D (the degree of the nonseasonal differencing polynomial); p is the order (number of time lags) of the autoregressive model; and q is the order of the moving-average model. The general form of an ARIMA (p,D,q) model is presented in Equation (1):(1)$$\Delta^{D}y_{t}=c+\phi_{1}\Delta^{D}y_{t-1}+\ldots +\phi_{p}\Delta^{D}y_{t-p}+\varepsilon_{t}+\theta_{1}\varepsilon_{t-1}+\ldots +\theta_{q}\varepsilon_{t-q}\;$$

In Equation (1), $\Delta^{D}y_{t}$ denotes a Dth differenced time series, and $\varepsilon_{t}$ is an uncorrelated innovation process with a mean of zero. The order D can be determined by stationarity tests, while the orders p and q can be informally determined by analyzing the autocorrelation or partial autocorrelation functions of the studied time series, or formally determined by using informational criteria. To fit the ARIMA model to the response data, the constrained maximum likelihood method was used, which generates maximum likelihood estimates with general parametric constraints (linear or nonlinear, equality or inequality), using the sequential quadratic programming method [18].

ARIMA models are time series analysis models used to better understand data or make future predictions. In order for the ARIMA models to correctly fit the data, the time series must be stationary, and even if it is not, stationarity can be achieved by transforming and differentiating the data. The utility of ARIMA models mainly lies in their ability to provide an estimate of the variability to be expected between future observations in function of past values and random errors [19].

Once the data partitions for the application of the ARIMA model are defined, the stationarity of the time series is verified. Thus, when data show a tendency to increase or decrease and have a certain pattern (seasonality), then they are not stationary. To statistically confirm the stationarity of the data, the augmented Dickey–Fuller (ADF) unit root test was used, according to Equation (2):(2)$$y_{t}=c+\delta t+\phi y_{t-1}+\beta_{1}\Delta y_{t-1}+\ldots +\beta_{p}\Delta y_{t-p}+\varepsilon_{t}\;$$

In Equation (2), c is a constant; δ is the coefficient on a time trend; p is the lag order of the autoregressive process; Δ is the differencing operator (${\Delta y}_{t}=y_{t}-y_{t-1}$); p is the number of lagged difference terms; and $\varepsilon_{t}$ is a mean zero innovation process. The ADF test informs the degree to which a null hypothesis (ϕ=1) that a unit root is present in the tested time series can be rejected or not to determine the stationarity of the data [20].

There are several methods that can be used to stagnate a series of data, including differentiation and transformation. In this work, the stationarity of the original series was verified, with no need to use the first differences. The parameters of the ARIMA model were not estimated from the graphs of the autocorrelation function (ACF) and the partial autocorrelation function (PACF), as there were no significant correlation points. Thus, the best model was selected considering the Akaike (AIC) and Schwarz, or Bayesian (BIC) [21] information criteria, which are likelihood-based measures of model fit that include a penalty for model complexity (the number of parameters). The AIC compares models from the perspective of information entropy as measured by Kullback–Leibler divergence. The BIC compares models from the perspective of decision theory, as measured by expected loss. The AIC and BIC for a given model are determined by Equations (3) and (4), respectively:


(3)
$$\mathrm{AIC}=-2\;log\;log\;L\left(\hat{\theta }\right)+2k$$



(4)
$$\mathrm{BIC}=-2\;log\;log\;L\left(\hat{\theta }\right)+\;k\;log\;log\;\left(T\right)\;$$


In Equations (3) and (4), $log\;log\;L\left(\hat{\theta }\right)\;$ denote the value of the maximized log likelihood objective function for a model with k parameters fit to T data points. To determine the order of the ARIMA(p,D,q) model, the terms p, D and q were tested in the range between 0 and 4, from ARIMA(0,0,0) to ARIMA(4,4,4), previously established based on the ACF and PACF graphs (resulting in 125 possible combinations). The one with the lowest value, in terms of AIC and BIC, was selected as the most suitable model for the time series of the study. The root mean squared error (RMSE) between the predicted values and the real values for all 125 ARIMA models tested was also verified [22,23].

To predict the dimension of new notifications of Hansen’s disease, some ARIMA or Box and Jenkins models were tested, that one to present the lowest values for the two information criteria AIC and BIC, and the one to best fit the data were used in the analysis. The Matlab R2021a software was used to perform the model fit.

This way, a predictive model was created for the notifications of new Hansen’s disease cases, and with this model, it was possible to identify the number of cases that were under-reported during the period from April to December 2020, comparing the prediction with the observed data for the pre-pandemic period (from April 2016 to March 2020).

## 3. Results

A total of 7035 notifications from 2001 to 2020 in the city of Palmas were analyzed. Table 1 shows the results of the indicators studied during the period and the proportion of cases in relation to gender and operational classification. The coefficient of general detection and in children under 15 years of age was more hyperendemic in 2018, with 270.68 and 19.19 cases/100,000 inhabitants, respectively. The proportion of multibacillary individuals was 98.99% and with a predominance of women (54.56%) in 2018.

The general detection indicators (Figure 2A) and those under 15 years of age (Figure 2B) showed an upward trend in the averages during the same periods. The proportion of cure in the cohort (Figure 3A) did not vary during the period, with a linear tendency. The proportion of cases diagnosed with grade 2 (Figure 3B) varied during the period, showing an increasing tendency.

Regarding the operational classification, from 2014 onwards, Figure 4A shows a decline in paucibacillary cases from 37.58% in 2014 to 3.05% in 2020; conversely, there was an increase in cases diagnosed as multibacillary from 62.42% to 96.95% during the same period. Until the year 2016, cases were more frequently reported in men, and after that year the frequency was higher in women, remaining very close until 2020, as seen in Figure 4B.

In the observation of the series on notifications grouped by month/year, two important changes in the averages during the period were identified, as shown in Figure 5, with the average for each segment being, respectively, 18.8525, 72.1042, and 28.2222.

The first significant change, which occurred at point 184 (April 2016), results from the realization of the Palmas Free from Hansen’s Disease Project [6]. From the actions realized during the aforementioned project, the trend would be to improve the reach and identification of people affected by the disease, ensuring that at least the notification indicators were maintained, which would establish a new standard in the series, if there was not an external factor. The second significant change was noticed at point 232 (April 2020), the month during which exponential growth in the number of confirmed cases of COVID-19 in Palmas was observed, when the population was guided towards isolation which, together with other factors, induced this change in the pattern of notifications of new cases of Hansen’s disease.

Due to these changes at the series level, it was decided to perform modeling using the first period between January 2001 and March 2016 for the pre-sampling, which contains data used to initialize lagged values in the model—and the second period, from April 2016 to March 2020, was used for model adjustment. Forecasts were made for the period between April and December 2020.

The stationarity of the series was confirmed by the augmented Dickey–Fuller (ADF) test, due to the null hypothesis (H0:ϕ=1) that the series is not stationary, which was rejected with a *p*-value = 0.001. The ARIMA (4,0,3) model was selected as the best model, i.e., the one that presented the lowest values for the two information criteria tested and that best fit the data (lowest RMSE between predicted and actual values for the training set), with AIC = 431.30 and BIC = 462.28, using the statistical significance level of 0.05. Figure 6 shows that the model closely follows the notifications observed during this period.

Comparing the predicted values with those existing in the database during the pandemic period (April to December 2020), it was possible to identify the presence of under-reporting, or hidden prevalence, with an average of 117% during this nine-month period (the reported values are, on average, 117% lower than those predicted by the proposed model). In terms of precision, the proposed model presented a difference of 0.57% between the actual and predicted mean values during the pre-pandemic period (48 months). In terms of absolute value, the average difference between actual and predicted values was 0.41 (the data range was between 37 and 141).

In the analysis of the autocorrelation of the residuals, as shown in Figure 7, it was confirmed that the model satisfies the assumption that the residuals are independent; that is, when there are no significant correlations (ACF and PACF plots), there may be one or two significant correlations in higher lags, due to random error, which does not mean that the assumption was not fulfilled. In this case, the standardized residuals plot is uncorrelated, showing randomly dispersed residuals around zero (averaging 0.0230), with constant variance, 81.25% concentrated between −2 and 2 and very few points (2.08%) above 3 or below −3, and the residuals distribution plot is approximately normally distributed, which are both conditions for an ARIMA model to be well adjusted [24].

## 4. Discussion

Hansen’s disease in Brazil is still a serious public health problem, as the country has high incidence rates, which makes it an area of very high endemicity [4]. In the state of Tocantins between 2014 and 2016, the municipality of Palmas identified an increase in the general detection indicator of 104.6% during the implementation of the Palmas Free from Hansen’s Disease Project, in addition to an increasing number of multibacillary cases, as shown in Figure 4A; in addition, from 2016 to 2019, the average number of notifications was 234/100,000 inhabitants, which maintained an increasing tendency. These data show the diagnostic difficulty, lack of trained professionals to identify cases, as well as the high hidden prevalence, which prove that the work of training health professionals and active searching carried out during the project were very effective, enabling primary care to realize an accurate recognition of leprosy [6].

Despite all the efforts of active searching and training for the detection and control of the disease, there was a reduction in the number of notifications by 54% from 2019 to 2020 with the arrival of the pandemic in Palmas. Unfortunately, this reduction in cases likely arose due to restrictions imposed by COVID-19. Factors such as isolation and social distancing caused many people to stop seeking care or have their treatment canceled. This reality was aggravated by the fact that leprosy is a chronic disease that requires early and uninterrupted treatment to prevent irreversible sequelae.

Visualizing the difficulties that would be found during this period, the Brazilian Society of Hanseology advised that the treatments be maintained despite the pandemic situation, evaluating specific issues such as the visits of supervised patients’ cases to health units and providing prescriptions for more than a month. It suggested that there should be an articulation of the coordination of the control programs of the municipalities and states to make these actions feasible, guaranteeing an extra supply of medicine. Additionally, there have been warnings that simultaneous infections of both Hansen’s disease—and thus its medication—as well as COVID-19 could cause liver problems and serious liver damage [25].

All this concern is intensified when one observes the indicator of the coefficient of detection in children under 15 years of age, one of the main epidemiological indicators, which indicates the prevalence of the disease with active transmission in the community. Hansen’s disease in children is a warning that requires immediate measures to break the chain of transmission. The challenge is to overcome all obstacles related to factors that influence the diagnosis of children and adolescents, such as stigma and prejudice, socioeconomic conditions, and difficulties in carrying out the examination in children. To reduce the disease in childhood, early diagnosis and the thorough examination of contacts are essential, as intimate intra-household contacts are the main transmitters [26,27,28].

Another study from 2001 to 2016 in Brazil observed an average detection coefficient in children under 15 years of age of 5.77/100,000 inhabitants, with Tocantins maintaining a stationary hyperendemic trend and Palmas showing a hyperendemic average, demonstrating permanence in transmissibility and the difficulty in the control of Hansen’s disease [29]. The study was confirmed when observing the period after the work was performed, during which the municipality of Palmas maintained the hyperendemic indicator even after the occurrence of the pandemic.

Fujishima’s [30] research correlated with notifications of Hansen’s disease in children under 15 years of age with socioeconomic information georeferenced by neighborhood, in Belem-Pará, another hyperendemic municipality. In this work, the correlation showed significance between notifications and lack of income, the absence of garbage collection, and the absence of sewage collection. Additionally, using spatial analysis, Barreto et al. [31] randomly tested children through serological assessment in public schools located in hyperendemic sectors, realizing that children diagnosed with the disease or with high anti-PGL-I serological concentration were located in places with a high concentration of the disease.

These studies emphasized the need to further research on high frequency clusters in the occurrence of notifications in Palmas, to then prioritize vulnerable areas using methodologies such as active search, not ignoring that the disease is related to other factors such as genetics and the low quality of healthcare and poor access to it. Identifying critical regions can increase the effectiveness of public health actions and reduce costs for disease control.

The Hansen’s disease detection coefficient in children under 15 indicates that it is hyperendemic, despite the decrease in notifications in 2020, together with the proportion of more than 95% of cases with multibacillary operational classification, which shows sources with high bacillary load, and with the increasing bias of reported cases with grade II. This scenario reflects the maintenance of hidden prevalence and late detection, concurrently with the pandemic situation, making it clear that there is still a long way to go to reach the World Health Organization (WHO) 2030 targets, which deal, among others, with the reduction in patients diagnosed with grade II [32].

In the municipality of Palmas, after the Free from Hansen’s Disease Project, there was significant progress in the detection and control of the disease. This evolution was possible through training in primary care health units and active search in the region [6,33]. This is evident when the first abrupt increase in the average of the notifications segment grouped by month/year is noticed (Figure 5), started together with the project. This average remained stable until April 2020, when the second change in the segment average occurred, from 72.104 to 28.222 average cases per month/year, due to the restrictions imposed by the pandemic.

In addition to the above problems, there are many difficulties related to access to health in a pandemic situation, such as the reduction in financial and human resources in national programs aiming to combating the disease, the cancellation of activity in order to accomplish early diagnosis (active search), the suspension of these services in health services, the difficulty of providing MDT in some locations, difficulty of treating disease reactions and the suspension of care for the prevention of disability and physical and psychosocial disabilities. In addition, there are still factors related to the situation of vulnerability of these people, which involve obstacles preventing patients travelling to health units. All these factors can lead to huge setbacks in efforts to reduce the disease worldwide, as without early diagnosis and proper treatment, the transmission advances and infection in children tends to increase [10].

In Palmas, despite all the work realized, the pandemic brought an imminent setback in the actions that had been carried out. According to the ARIMA predictive model, the sum of notifications from April 2020 to December 2020 would be approximately 663.16 cases, but there were 254 cases observed, which indicates under-reporting and an increase in the hidden prevalence of the disease. This result shows the urgent need to restart the active search, strengthen searches, and contact examinations, as well as information and support programs for people with suspicion or affected by leprosy.

When analyzing the results, it should be considered that this is a quantitative study that is limited to the analysis of data recorded in the Unified Health System database, referring to the city of Palmas, TO. It is important to note that climatic, social, cultural characteristics, among other factors, can influence the conclusions of research carried out in other regions. Another limitation of the research would be related to the error margin of the predictive model; however, the results show consistency when analyzing the history of the endemic disease and other related works.

Thus, the results of this work emphasize the imminent need for public actions and policies aimed at the health of people affected by Hansen’s disease with the objective of improving the quality of life and care of services, which should be associated with the strict surveillance of their contacts, especially in children under 15 years, to prevent deformities in children. Another extremely relevant factor refers to the continuing education of health professionals, and not just in terms of theory, because the diagnosis is essentially clinical.

## 5. Conclusions

In the context of the pandemic, the weakness in the containment of Hansen’s Disease by the control programs has become more evident, and the lack or misunderstanding of information, the lack of minimum hygiene conditions (drinking water and soap), food and protective equipment such as masks, concomitant with the impossibility of formal work to ensure their survival in this scenario, placed people affected by this disease in a situation of greater vulnerability.

The ARIMA model reports that an average of 177% of new cases of Hansen’s disease in Palmas were not reported during the period of the COVID-19 pandemic in 2020. Moreover, in 2019, 236.65 cases were reported per 100,000 inhabitants, and of these, 96.95% were classified as multibacillary, that is, people with a high bacillary load that keep transmitting the disease to individuals they live with, which has a tendency to exponentially increase the number of people infected. Thus, this study shows the need for immediate actions by the municipal and state control program to actively search for cases and reduce hidden prevalence.

## Figures and Tables

**Figure 1 ijerph-19-00415-f001:**
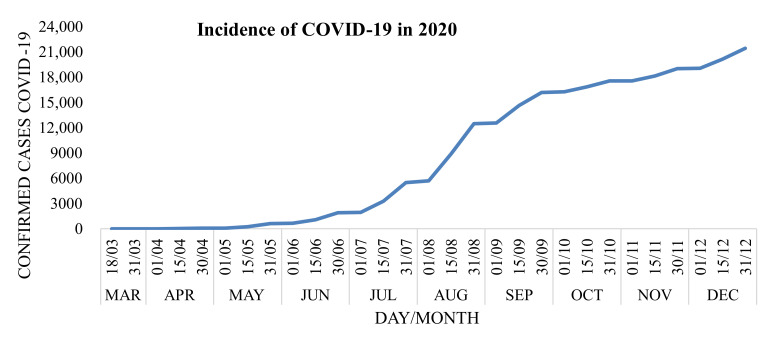
Incidence of COVID-19 in Palmas during the year 2020.

**Figure 2 ijerph-19-00415-f002:**
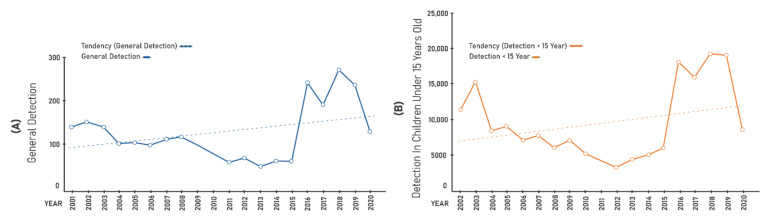
Hansen’s disease indicators (per 100,000 inhabitants) in Palmas, Tocantins, Brazil, during the period 2001–2020: (**A**) general detection; and (**B**) detection in children under 15 years old.

**Figure 3 ijerph-19-00415-f003:**
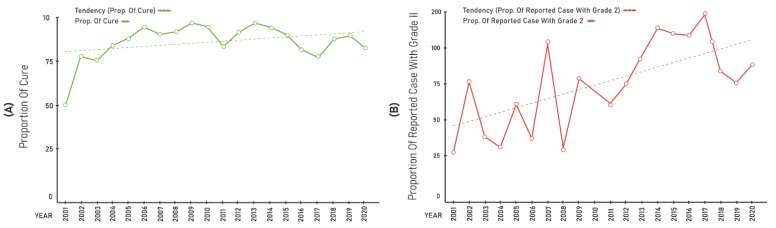
Hansen’s disease indicators (per 100,000 inhabitants) in Palmas, Tocantins, Brazil, during the period 2001–2020: (**A**) proportion of cure; and (**B**) proportion of reported cases with grade II.

**Figure 4 ijerph-19-00415-f004:**
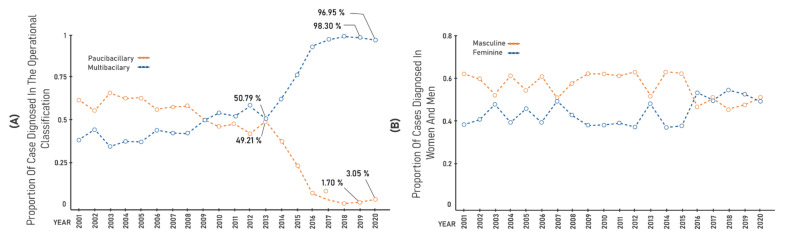
Proportion of new Hansen’s disease cases detected by operational classification and sex in Palmas, Tocantins, Brazil, during the period 2001–2020: (**A**) proportion of cases diagnosed in the multibacillary (MB) and paucibacillary (PB) operational classification; and (**B**) proportion of cases diagnosed in women and men.

**Figure 5 ijerph-19-00415-f005:**
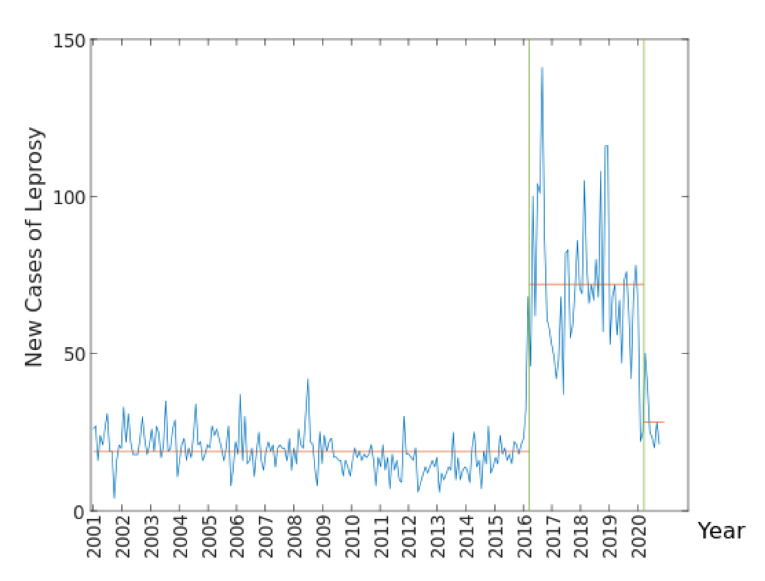
Spots of change in the averages of the series of notifications of new Hansen’s disease cases during the period 2001–2020 in Palmas, Tocantins.

**Figure 6 ijerph-19-00415-f006:**
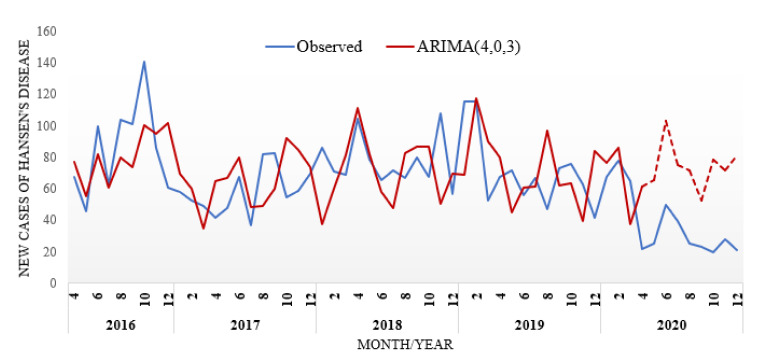
Notifications grouped by month/year: observed notifications and forecast with the ARIMA model (4,0,3).

**Figure 7 ijerph-19-00415-f007:**
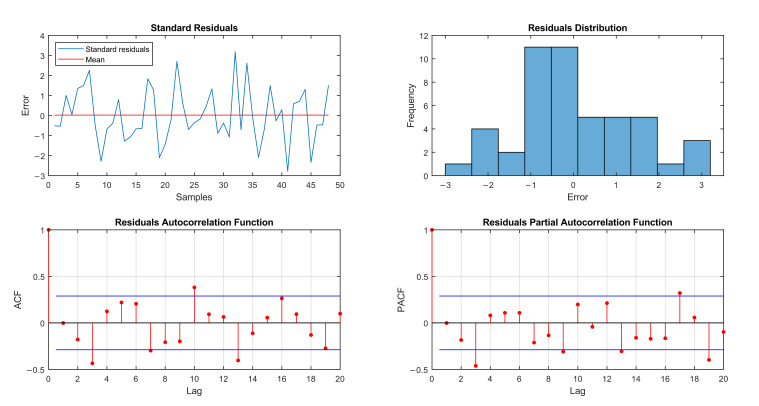
Residual analysis of the ARIMA model (4,0,3) applied to notifications of new cases of Hansen’s disease in the city of Palmas, Tocantins, from 2001 to 2020.

**Table 1 ijerph-19-00415-t001:** Epidemiological and operational indicators of Hansen’s disease during the period 2001–2020 in Palmas, Tocantins.

Years	Gen.Det	<15 Years	Cure	Grade II	% PB	% MB	% F	% M
2001	139.18	18.56	50.00	2.20	62%	38%	38%	62%
2002	150.80	11.23	77.68	6.13	56%	44%	40%	60%
2003	138.81	15.31	75.18	3.07	66%	34%	48%	52%
2004	101.26	8.37	83.87	2.46	63%	37%	39%	61%
2005	102.80	9.02	87.88	4.88	63%	37%	46%	54%
2006	97.33	7.07	94.29	2.95	56%	44%	39%	61%
2007	109.87	7.73	90.34	8.26	58%	42%	49%	51%
2008	115.75	6.02	91.59	2.48	58%	42%	42%	58%
2009	98.07	7.12	96.58	6.28	51%	49%	38%	62%
2010	77.08	5.15	94.69	5.58	46%	54%	38%	62%
2011	57.79	4.15	83.62	4.91	48%	52%	39%	61%
2012	66.92	3.22	91.21	5.98	42%	58%	37%	63%
2013	48.86	4.30	96.55	7.59	49%	51%	48%	52%
2014	59.15	4.94	94.05	9.09	38%	62%	37%	63%
2015	59.40	5.91	89.80	8.80	23%	77%	38%	62%
2016	241.20	18.02	81.11	8.70	7%	93%	53%	47%
2017	189.34	15.82	77.30	9.84	3%	97%	50%	50%
2018	270.68	19.19	87.45	6.70	1%	99%	55%	45%
2019	236.35	19.06	89.23	6.06	2%	98%	53%	47%
2020	128.31	8.49	82.40	7.06	3%	97%	49%	51%

Legend: Gen.Det—annual detection coefficient of new cases per 100,000 inhabitants; <15 year—annual detection coefficient of new cases in people aged 0–14 years per 100,000 inhabitants; Cure—proportion of cured cases among new cases in the cohort years; Grade II—proportion of cases with grade II physical disability among the new cases detected in the year; % PB—proportion of notifications of new paucibacillary cases; % MB—proportion of notifications of new multibacillary cases; % F—proportion of notifications of new cases by female sex; % M—proportion of notifications of new cases by male sex.

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
