# Peer review of "Application of the ARIMA Model to Predict Under-Reporting of New Cases of Hansen’s Disease during the COVID-19 Pandemic in a Municipality of the Amazon Region"

_ijerph, 2021, doi:10.3390/ijerph19010415_

Round 1

Reviewer 1 Report

--Review comments for manuscript ID: "ijerph-1464567-peer-review-v1", entitled "Application of the Arima Model to Predict Underreporting of New Cases of Hansen's Disease During the Covid-19 Pandemic in a Municipality of the Amazon Region" of the Journal of “International Journal of Environmental Research and Public Health".

--General:

Cunha et al. conducted a modeling analysis by employing an Arima Model to Predict Underreporting of New Cases of Hansen's Disease During the Covid-19 Pandemic in a Municipality of the Amazon Region. The ARIMA model reports that 177% of new cases of Hansen’s disease were not reported in Palmas during the period of the Covid-19 pandemic in 2020. The study shows the need for immediate actions by the municipal control program for the active search for cases and reduction of hidden prevalence.

This is a nice and well-conducted modelling study, which addressed an epidemiological problem of timely and public health importance by assessing the impact of the COVID-19 pandemic on Hansen’s disease. I consider their method reasonable and sophisticated, and the results are presented logically and consistently. The analyzing outcomes support the main results with biologically and clinically appropriate settings. Thus, I would recommend it for publishing after the following concerns are considered.

--Comments:

--I wonder what is the prediction accuracy? Please, clarify.

--The procedure of how the ARIMA model was estimated should be explicitly stated in the method section. Also, the version of the software used should be stated. That is, the authors need to describe in more detail how they develop models, especially the ARIMA. How to determine the order of ARIMA was undescribed clearly.

--How can your model provide any policy implication or prediction for the COVID-19 control?

--In the first paragraph of the introduction, I suggest the authors cite the following two websites (WHO and CDC). https://www.who.int/news-room/fact-sheets/detail/leprosy, https://www.cdc.gov/leprosy/index.html.

--The sentence “The existing difficulties in access and information of health are aggravated and, along with the cancellation of early diagnosis activities, suspension of care and some specific problems in the provision of multidrug therapy (MDT), in addition to factors related to the vulnerabilities of these patients, can cause a huge setback in efforts to reduce the burden of this disease in the country.” need a proper citation(s).

--I appreciate the authors have done a very nice discussion, mainly focusing on the technical part. I would be more appreciative if the author could elaborate on the epidemiology and public health sides.

--Limitations of the study should be clearly stated in the discussion section. Also, epidemiological/clinical implication(s) of the results should be discussed in the introduction part.

--Table 1 entries are confusing, I suggest the authors write them in full and in the English language as well as explain each of them in the caption.

--I suggest the authors should provide the epidemiological background of COVID-19 in Palmas, Tocantins, Brazil, in the introduction section, as done for the leprosy disease.

Reviewer 2 Report

Valéria Perim da Cunha et al. use the ARIMA model to predict underreporting of new Hansen’s disease cases during the Covid-19 pandemic in Palmas, Tocantins, Brazil. Based on the model, they conclude that 177% of new cases of Hansen’s disease were not reported in Palmas during the period of the Covid-19 pandemic in 2020.

The introduction of the paper is really well written and organized. One part that I think can be greatly improved is in the presentation of the results, and the decision-making that leads to the choice of the ARIMA (4,0,3) model. If the author can present other models that were being considered, and the decision that led the authors to choose the (4,0,3) model and reject the other models, it will make the paper much more convincing.

It would also be great if the authors can discuss further the significance of Figure 2A. I would be interested to know what the author's thought is on why the Paucibacillary fraction decreases after 2014.

Also, for Figure 5, I am not clear what the purpose of the top two graphs are. If the author can explain the graphs better in the Results or Discussion section, that will be great!

Reviewer 3 Report

Thank you for this study, which investigates the ARIMA models as a method to detect case under-reporting for Hansen's Disease, during the COVID-19 pandemic. Some comments to improve the manuscript follow:

Can the labels in Table 1 be presented in English - with appropriate definitions provided in the caption. In general, the Table and Figure captions require more information. Some font sizes are too small, and as such, it was difficult to read and understand the results.

Materials and Methods - can some example formulae please be included, for ARIMA and  Augmented Dickey-Fuller (ADF). What features provide an advantage for the detection of under-reporting? The "Predictive Model" (2.4) section carries much description etc that would be better suited to the Discussion section; please condense this section, and as suggested above, provide more statistical details.

In general - It wasn't clear to me what advantages are provided by ARIMA? And how did you valaidate that there was under-reporting? (namely, the 177% over estimate of reported cases). Is retrospective testing of samples possible, or access to other data? This is the primary claim of this research study, and requires more explanation and context. You clearly stated the health and hygiene issues that result in the high prevalance of Hansen's Disease, and some public health advice, as informed by your ARIMA results, would be a nice addition to the Discussion.

Round 2

Reviewer 1 Report

---R1

--Review comments for manuscript ID: "ijerph-1464567-peer-review-v1", entitled "Application of the Arima Model to Predict Underreporting of New Cases of Hansen's Disease During the Covid-19 Pandemic in a Municipality of the Amazon Region" of the Journal of “International Journal of Environmental Research and Public Health".

---

General:

I appreciate that the authors have carefully addressed all the issues raised by the reviewers that might improve the manuscript from its initial draft. I think the paper is recommended for publication after correcting some typos throughout the manuscript. Also, there is no need of numbering the keywords “Leprosy 1; Neglected disease 2; Covid-19 3; Time Series Analysis 4; Epidemiology 5”, numbers should be removed.

Note that, there is no need for further revision as the manuscript has improved significantly.

Reviewer 2 Report

For this second revision of the paper, Valéria Perim da Cunha et al improves on the previous manuscript. My comments have been addressed and the paper should be ready for publication after minor corrections. Specifically, there are some grammatical and wording change that would improve the paper. Here are a few I would suggest:

  • Figure 1: change 'evolution of COVID-19' to 'incidence of COVID-19'
  • Page 2 last paragraph, perhaps change to: 'In Palmas, at the end of the first year of the pandemic, there is a decrease in the notifications of new cases of leprosy'
  • Page 3 first paragraph, perhaps change to: '...providing information on the evolution of the number of leprosy cases as a result of the pandemic...'
  • Page 5, the paragraph after equation 4: 'maximized loglikelihood' to 'maximized log likelihood'
  • Page 10, last paragraph. Perhaps change to: 'Unfortunately, this reduction in cases likely arose due to restrictions imposed by COVID. Factors such as isolation and social distancing caused many people to stop seeking care or had their treatment canceled. '
  • Page 12, 4th paragraph. Perhaps change to: 'this is a quantitative study that is limited to the analysis of data recorded in the Unified Health System database'
